# Treatment Outcomes of Proton Beam Therapy for Esophageal Squamous Cell Carcinoma at a Single Institute

**DOI:** 10.3390/cancers15235524

**Published:** 2023-11-22

**Authors:** Eun Sang Oh, Sung Ho Moon, Youngjoo Lee, Beung-Chul Ahn, Jong Yeul Lee, Yang-Gun Suh, Joo-Hyun Chung, Moon Soo Kim, Jong Mog Lee, Jin-Ho Choi, Tae Hyun Kim

**Affiliations:** 1Proton Therapy Center, Research Institute and Hospital, National Cancer Center, Goyang 410-769, Republic of Korea; euns0530@ncc.re.kr (E.S.O.); suhmd@ncc.re.kr (Y.-G.S.); gag920@ncc.re.kr (J.-H.C.); k2onco@ncc.re.kr (T.H.K.); 2Department of Internal Medicine, Research Institute and Hospital, National Cancer Center, Goyang 410-769, Republic of Korea; yjlee@ncc.re.kr (Y.L.); abcduke@ncc.re.kr (B.-C.A.); 3Center for Gastric Cancer, Research Institute and Hospital, National Cancer Center, Goyang 410-769, Republic of Korea; jylee@ncc.re.kr; 4Center for Lung Cancer, Department of Thoracic Surgery, Research Institute and Hospital, National Cancer Center, Goyang 410-769, Republic of Korea; vsd10@ncc.re.kr (M.S.K.); jongmog@ncc.re.kr (J.M.L.);

**Keywords:** esophageal carcinoma, squamous cell carcinoma, radiation therapy, proton beam therapy

## Abstract

**Simple Summary:**

This study investigated the effectiveness of proton beam therapy (PBT) for esophageal squamous cell carcinoma (ESCC). This research found that PBT showed promising results in terms of favorable overall survival rates and reducing toxicities in ESCC patients. The 3 year overall survival rates for patients with stages I, II, and III of ESCC were 81.0%, 62.9%, and 51.3%, with corresponding progression-free survival rates of 70.6%, 71.8%, and 39.8%. Notably, salvage procedures were successful at treating isolated local and regional progression, and severe lymphopenia cases were absent. This study supports the conclusion that PBT is an effective treatment option for ESCC patients in terms of both the survival outcomes and toxicity management.

**Abstract:**

Recently, proton beam therapy (PBT) has gathered attention for improving outcomes and reducing toxicities in various cancers; however, the evidence for esophageal squamous cell carcinoma (ESCC) is lacking. Our study retrospectively evaluated the outcomes of PBT for ESCC patients at a single institute. The patients treated with PBT between November 2015 and February 2022 were included in the study, excluding those with distant metastases or those that had undertaken prior treatment for esophageal cancer (EC). The 3 year overall survival (OS) and progression-free survival (PFS) rates were calculated based on stage grouping. The patterns of failure, salvage treatment outcomes, and toxicity profiles were analyzed. The median follow-up was 35.1 months, and 132 patients were analyzed. The 3 year OS and PFS rates for the stages I, II, and III disease cases were 81.0%, 62.9%, and 51.3%; and 70.6%, 71.8%, and 39.8%, respectively. Nineteen patients presented isolated local progression, ten patients underwent appropriate salvage procedures, and nine were successfully salvaged. One patient with isolated regional progression was also salvaged. No cases of grade ≥ 4 lymphopenia were observed. One patient had grade 4 pericardial effusion and esophageal fistula. For the patients with ESCC, PBT is an effective treatment in terms of the survival outcomes and toxicities.

## 1. Introduction

Esophageal cancer (EC) has a high fatality rate, ranking sixth in terms of cancer mortality worldwide [1], and its incidence rate is continuously rising. Although the incidence of esophageal adenocarcinoma (EAC) is increasing in Western countries, the main histologic type of ECs is esophageal squamous cell carcinoma (ESCC), accounting for 84% of the cases worldwide, being the most prevalent subtype of EC in East Asia [2]. However, the clinical research on patients with pure ESCC without EAC is insufficient [3,4].

For decades, various multidisciplinary approaches combining surgery, radiotherapy (RT), and chemotherapy have been used to improve the treatment outcomes of EC. Neoadjuvant chemoradiotherapy (nCRT) followed by surgery has become the standard of care for operable ECs since a series of landmark clinical trials reported successful results [5,6]. Furthermore, active surveillance rather than immediate surgery for patients with locally advanced EC with a complete clinical response who cannot tolerate or do not want surgery has been gathering attention, as the pathological complete response rate after nCRT for ESCCs is reportedly as high as 50.0–70.6% [3,7].

Recent advances in RT technology, such as intensity-modulated RT (IMRT) and proton beam therapy (PBT), ensure the delivery of adequate radiation doses to the tumor, while limiting the doses to critical organs at risk, such as the spinal cord, lungs, and heart. To date, dosimetric and clinical studies have shown superior cardiopulmonary dose sparing results, possibly leading to less toxicity and the preservation of circulating lymphocytes, which are critical in PBT’s anti-tumor response, compared to those of three-dimensional conformal radiotherapy (3D-CRT) and IMRT [8,9,10]. However, clinical studies on PBT for EC, especially for pure ESCCs, are still lacking. Therefore, the aim of this study was to evaluate the treatment outcomes of definitive PBT as the sole radiation modality with or without chemotherapy in thoracic ESCC patients at a single institute.

## 2. Materials and Methods

### 2.1. Patients

Patients treated with PBT for ESCC between November 2015 and February 2022 were retrospectively reviewed. All the patients were histologically confirmed as having ESCC prior to treatment and were assessed for their clinical stage using esophagoscopy, endoscopic ultrasonography, chest computed tomography (CT), and 18F-FDG positron emission tomography (PET)/CT scan; staging was determined according to the American Joint Committee on Cancer TNM staging system [11]. Demographic and clinical information were collected from medical records, including sex, age, Eastern Cooperative Oncology Group performance score (ECOG PS), and disease progression sites and dates. Such data were anonymized after assigning case numbers. The patients without metastases to distant organs, no other sites with uncontrolled cancer within 2 years before the treatment, and no other prior treatments for EC were included in this study, and 132 patients were analyzed. This study was approved by the Institutional Review Board (IRB number: NCC2023-0273) of the NCC.

### 2.2. PBT Planning and Delivery

All the patients were positioned on a round-type couch in an arm-up, supine position and immobilized using a vacuum cushion. As mentioned in a previous paper, 3D-based passive scattering PBT planning with an appropriate planning target volume (PTV) margin considering the set-up error, respiration, and range uncertainty was used in the earlier period of this study [12]. Subsequently, a 4D-CT-based treatment simulation after assessing the patient’s breathing pattern was used if possible; in cases with an irregular pattern, an abdominal compression belt was applied. To perform setup verification, we used weekly cone-beam CT and daily X-ray techniques. Generally, a conventional fractionation of 2.0 Gy of cobalt gray equivalent (CGE) of PBT was delivered once daily five times per week, and the total dose had a range of 45–70.2 CGE (median, 66 CGE). One patient who received 45 CGE was initially planned to receive a treatment up to 63 CGE, but the treatment was discontinued due to their persistent poor general condition. All the other patients received a radiation dose of 54 CGE or higher.

The gross tumor volume (GTV) was defined as the primary tumor (GTVp) and metastatic lymph node volumes (GTVn) visualized on the chest CT and PET/CT scans. For the patients who received concurrent chemotherapy, the clinical target volume (CTV) was extended by 3.5–4 cm craniocaudally, 0.5–1 cm circumferentially from GTVp, and 0.7–1 cm circumferentially from GTVn. For EFRT, the elective CTV encompassed the entire esophagus and regional LN group, including the pretracheal, retrotracheal, paratracheal, subcarinal, and peri-esophageal LNs with or without the supraclavicular LN (SCN) and a part of the abdomen, including the paracardial, left gastric, and celiac LNs, as described in detail elsewhere [13]. The planning target volume was defined as the CTV plus 0.5–1 cm.

PBT plans were produced using the EclipseR planning system (Varian Medical System, Palo Alto, CA, USA), and passive-scattering or pencil-beam scanning (PBS) techniques were used. For PBS planning, field-specific PTVs were generated for each field, and maximal efforts were made to adjust the range uncertainty with multi-field robust optimization.

### 2.3. Chemotherapy

Concurrent chemotherapy was administered to all the patients with stages II-III and some with stage I disease. A clinical trial conducted at our institution involved treating the majority of the stage I patients without applying chemotherapy and utilizing EFRT. However, in the stage I cases deemed unfavorable for RT alone, such as a long-segment primary tumor spread or small, but suspicious, lymph node metastasis, a multidisciplinary team may decide to combine RT with chemotherapy in a tumor board meeting. Additionally, there might be slight variations in decision making among the radiation oncologists at our institution when it comes to treating EC. The selection of the chemotherapy regimen is decided at the discretion of the medical oncologists. The most commonly used regimen was four–six cycles of weekly intravenous carboplatin (AUC of 2) and paclitaxel (50 mg/m^2^). The other regimens used were capecitabine with or without platinum and bolus 5-fluorouracil (5-FU) regimens, as follows: capecitabine was taken orally twice daily for 14 days at a dose of 2500 mg/m^2^/day; 5-FU was taken at 600–700 mg/m^2^/day with a 3 week interval; and cisplatin was infused intravenously at 60–75 mg/m^2^ with a 3 week interval.

### 2.4. Clinical Assessment

The patients were followed up at 3–6 months intervals via chest CT and esophagoscopy at least twice a year. Local and regional progression were defined as regrowth or progression of the primary tumor or the development of metachronous primary EC and regional lymph node (LN) metastasis, respectively. Distant LN progression and solid organ metastasis, such as to the liver or lungs, were separately defined. The survival duration was calculated from the date of RT initiation until the date of the last follow-up or the occurrence of events, such as any type of progression or death. Acute toxicity was defined as the toxicity emerging within three months from the initiation to completion of treatment, and at subsequent time points, toxicity was defined as late toxicity. Toxicities were analyzed based on the Common Terminology Criteria for Adverse Events (version 5.0).

### 2.5. Statistical Analysis

Statistical analyses were performed using IBM SPSS Statistics Subscription (IBM, Armonk, NY, USA) and Excel (Microsoft, Redmond, WA, USA). Kaplan–Meier survival curves were used to estimate the survival outcomes. The log-rank test was used to compare the survival differences in univariate analysis, and a stepwise forward selection procedure was used in multivariate analysis. Hazard ratios (HRs) were estimated using the Cox proportional hazards model. Statistical significance was set at *p* < 0.05.

## 3. Results

### 3.1. Patients’ Characteristics

A total of 132 patients with a median age of 70 years (range, 40–89 years) were included in this study. The tumor location was defined according to the epicenter of the tumor, and most of the tumors were located in the middle (n = 53, 40.2%) and lower (n = 59, 44.7%) thoracic areas. All the patients received PBT as the sole radiation modality, with 76 (57.6%) patients undergoing passive scattering and 56 (42.4%) PBS. For the patients with stage I EC that underwent EFRT (n = 81, 61.4%), chemotherapy was not administered, but a higher radiation dose with 60–70.2 CGE was prescribed according to our institutional policy. For the patients who received concurrent chemotherapy, the most commonly used regimen was intravenous carboplatin and paclitaxel. The baseline patient and tumor characteristics are summarized in Table 1.

### 3.2. Survival Outcomes and Patterns of Disease Progression

The median follow-up duration was 35.1 months (range, 1.6–85.0 months). The 3 year overall survival (OS) and progression-free survival (PFS) rates were 73.1% and 66.2%, respectively (Figure 1A,B). The 3 year OS and PFS rates for the patients with stage I, II, and III diseases were 81.0%, 62.9%, and 51.3%; and 70.6%, 71.8%, and 39.8%, respectively (Figure 1C,D). During the follow-up period, 37 deaths were observed. The causes of death were as follows: disease progression (n = 28), septic shock after salvage esophagectomy for regional progression (n = 1), other malignancies (n = 4), an underlying lung disease (n = 3), and unknown (n = 1).

The patterns of disease progression are summarized in Figure 2. At the time of analysis, 89 patients (66.9%) showed no evidence of disease progression. Local, regional, and distant progression developed in 28 (21.2%), 15 (11.3%), and 14 patients (10.6%), respectively, and the actual 3 year local and loco-regional control rates were 76.6% and 68.3%, respectively. Nineteen patients presented isolated local progression, eighteen experienced primary tumor progression, and one had metachronous primary EC. Sixteen of them (84.2%) were considered salvage candidates; however, nine of them did not undergo salvage procedures due to them having a poor general condition (n = 3) and refusing further treatment (n = 6). Ten patients underwent a suitable salvage procedure. Among the sixteen patients, nine (47.3%) were successfully salvaged. For those nine patients who were successfully salvaged, four underwent surgery, four received endoscopic submucosal dissection, and one was treated with argon plasma coagulation. Six patients presented isolated regional progression, with two identified as salvage candidates. Among them, two underwent a surgical salvage procedure, resulting in one successful salvage.

### 3.3. Prognostic Factor Analysis

In the univariate analysis of OS, the ECOG PS, and cT, and cN categories were identified as significant prognostic factors, while the presence of concurrent chemotherapy or undergoing a PBT technique did not show a significant difference (Appendix A). In the multivariate analysis, after adjusting for the other covariates, only the cN category (HR = 5.442, CI 2.167–13.664, *p* = 0.000) was observed as a significant prognostic factor for OS (Appendix A).

### 3.4. Treatment-Related Toxicities

The adverse events observed in our study are summarized in Table 2. Regarding acute toxicities, three patients (2.3%) presented with grade 3 esophagitis, and about half of the patients presented with lymphopenia. However, none of the patients presented with grade ≥ 4 of such complications.

Although late toxicities did not occur in most patients, grades 1–2 pleural effusion were relatively common (n = 34, 25.8%), and grades 1–2 pericardial effusion and esophageal stenosis were also observed in some cases. In our study, two patients experienced grades 3 and 4 esophageal fistulas. One patient initially required an esophageal stent due to T3 ESCC, and a fistula developed after chemoradiation. The other patient suffered from persistent esophageal edema with ulceration during and after undergoing PBT alone for a 5 cm whole circumferential segment tumor bed following endoscopic submucosal dissection. He was hospitalized and underwent percutaneous radiologic gastrostomy (PRG) insertion; subsequently, esophageal fistula infection was detected and progressed, and eventually, cardiac tamponade with a large amount of pericardial effusion was observed. Under delicate supportive medical care, the fistula and cardiac complications were resolved; therefore, we categorized the complications as a grade 4 pericardial effusion and esophageal fistula.

## 4. Discussion

The development of novel RT techniques has improved the survival outcomes and reduced toxicity of EC treatment, which is similar to other carcinomas. However, the actual efficacy of these theoretical benefits in a clinical setting remains pivotal. Particle radiation therapy is currently gathering attention, and while some studies report insignificant gains, others demonstrate improved survival outcomes [14,15]. The incidence of ESCC is increasing worldwide as a serious health issue. Although particle therapy reduces the toxicity in EC [10,14,16], a notable difference exists depending on histological subtypes. Therefore, these results should be carefully interpreted in the context of ESCC, which presents different characteristics and development patterns than EAC. Recently, a multi-institutional study in Japan retrospectively evaluated PBT for EC, in which most of the patients had ESCC; however, they applied PBT either as the sole modality or as a combination form X-ray RT [17]. Conversely, we employed PBT as the sole RT modality for patients with ESCC, which distinguishes our study from the others.

Overall, the survival outcomes of this study seem promising, with 3 year OS and PFS rates of 73.1% and 66.2%, respectively, which are similar to those of the previously mentioned Japanese multi-institutional PBT study [17]. When comparing the stages, the 3 year OS and PFS rates of 81% and 70.6%, respectively, in the patients with stage I disease of our study were comparable to those of the X-ray series (60.6–80.5%) [18,19,20] and to the results of the aforementioned Japanese study [17]. Notably, our study had a relatively higher proportion of patients with stage I disease, which can be attributed to the increasing trend of the early detection of EC during screening endoscopy. The issue of elective nodal irradiation (ENI) remains controversial, particularly in cases of superficial ECs, as approximately 20–25% of them present clinically undetectable LN metastasis [21]. Most cases of isolated local progression after chemoradiation can often be successfully salvaged, which is consistent with the findings of more than 80% that were deemed as local salvage candidates and the approximate 50% actual salvage rate in our study. However, the salvage of LN metastasis is challenging [22]. As per our institutional policy, EFRT-PBT without chemotherapy is recommended for selected patients with superficial EC to reduce the long-term complications by significantly decreasing the unnecessary radiation doses to critical organs surrounding the esophagus, while still aiming to achieve long-term survival outcomes [23]. Therefore, differences in the pattern of treatment failure were observed, especially when compared to the studies without ENI. In our study, the use of EFRT-PBT without chemotherapy showed promising results in maintaining relatively low rates of regional recurrence and distant metastasis. Additionally, the addition of chemotherapy in the stage I ESCC cases did not significantly impact the efficacy of the treatment outcomes. Regarding the stage II-III cases, the 3 year OS and PFS rates of our study were also comparable to those of the historical studies, which were reported to be 44.7–61.9% in both the X-ray and PBT series [7,15].

Previous PBT series for EC have demonstrated the dosimetric advantages of PBT over photon RT for sparing the cardiopulmonary system both in neoadjuvant and definitive treatment settings [8,9,14,16]. In our study, grade ≥ 3 acute or late lung and heart toxicities were rare, which is consistent with the findings of a Japanese multi-institutional study on PBT for EC (17). Moreover, the incidence of such toxicities in our study was much lower than those of the previous X-ray studies, which reported rates of 6.9–16% for grade ≥ 3 and 3.3–4.0% for grade ≥ 4 toxicities [24,25]. These results suggest that PBT may offer a favorable toxicity profile and potentially reduce the risk of severe lung and heart complications in patients with EC undergoing RT. However, although such dosimetric benefits have been observed, they do not always translate into significant clinical benefits for the patients [9]. The other factors, such as the tumor characteristics, patient comorbidities, and overall treatment approach, including the RT technique employed, can also impact the toxicity outcomes. We noted that grade 1–2 pleural and pericardial effusions, as well as esophageal stenosis, remain common issues in the era of PBT for EC. Therefore, making every effort to minimize these toxicities and explore strategies for their reduction is crucial.

From a hematological toxicity perspective, multiple studies have reported that PBT demonstrates a larger lymphocyte-sparing effect compared to that of photon-based RT. These studies have shown that the incidence of grade 4 lymphopenia, which has been correlated with a poor survival in various malignancies [26,27], is significantly lower in the patients treated with PBT (17.6–22.0%) than it is in those treated with photon-based RT (40.4–56.0%) [28,29]. This lymphocyte-sparing effect is of particular importance, as lymphopenia can impact immune function and the overall treatment outcomes. In our study, no cases of grade ≥ 4 lymphopenia were observed, and grade 3 lymphopenia was identified in only 22.0% of the patients. These findings are favorable compared to the results of previous studies, suggesting that our treatment approach with PBT may lead to better clinical outcomes. However, it should be considered that the tumor location in our study differed from that in the Western PBT series, as we had a similar proportion of tumors located in the middle and thoracic regions. Additionally, it should be noted that induction chemotherapy was not employed in our study. Considering the correlation between lymphopenia and the treatment response [26,27], the lymphocyte-sparing effect of PBT may be particularly beneficial, especially in combination with immunotherapies such as nivolumab, as demonstrated by the positive results of the Checkmate 577 trial in EC [4].

This study had several limitations that must be acknowledged. First, it was a retrospective analysis conducted in a single institute, which may have introduced a selection bias. However, the strength of this single-institute study is the homogeneity in the treatment regimens, including dose-fractionation, radiation field, treatment planning, and delivery; therefore, the obtained results can be considered reliable. Additionally, the retrospective nature of the study may have resulted in the underestimation of the treatment-related toxicities, as they may not have been fully documented in the medical records. To mitigate this limitation, regular follow-ups with chest CT and esophagoscopy were conducted to supplement the information related to late toxicity. Second, the distribution of disease stages within the patients in this study was skewed towards stage I, accounting for 70% of the cases. This is primarily due to the recent trend of the early detection of EC through esophagoduodenoscopy screening. Despite the variation in the number of patients in each stage, the treatment results were considered reliable, and consistent treatment was administered.

This study is the first to report the treatment outcomes of definitive PBT as the sole radiation modality for predominantly thoracic EC with a specific focus on pure ESCCs. Therefore, this study holds a significant value as it applies a uniform treatment regimen using PBT to a homogeneous patient population.

## 5. Conclusions

For patients with ESCC, PBT is an effective treatment modality in terms of both the survival outcomes and toxicities.

## Figures and Tables

**Figure 1 cancers-15-05524-f001:**
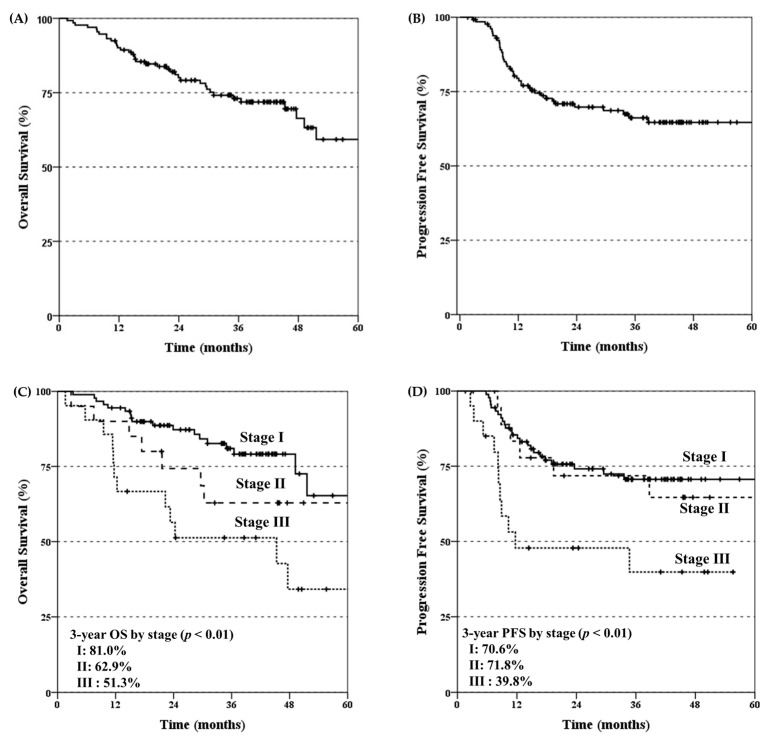
Overall survival (OS) and progression free survival (PFS) rates for all patients (**A**,**B**) and according to stage (**C**,**D**).

**Figure 2 cancers-15-05524-f002:**
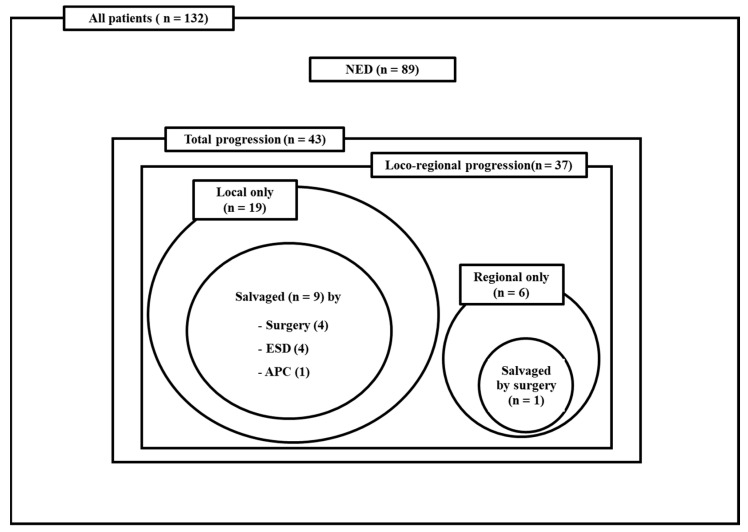
Patterns of disease progression and details of salvage procedures. Abbreviations: Local, local progression; Regional, regional progression; Distant, distant progression; NED, no evidence of disease progression, ESD, endoscopic submucosal dissection; APC, argon plasma coagulation.

**Table 1 cancers-15-05524-t001:** Patient characteristics.

Characteristics		Total, N (%)
Sex	Male	121 (91.7)
	Female	11 (8.3)
Age (year)	Median (range)	70 (40–89)
ECOG PS	0	62 (47.0)
	1	65 (49.2)
	2	5 (3.8)
Histology	Squamous cell carcinoma	132 (100)
Tumor location	Upper thoracic	16 (12.1)
	Middle thoracic	53 (40.2)
	Lower thoracic	59 (44.7)
	EG Junction	4 (3.0)
cT classification	T1a	16 (12.1)
	T1b	75 (56.8)
	T2	18 (13.6)
	T3	23 (17.4)
cN classification	N0	95 (72.0)
	N1	29 (22.0)
	N2	8 (6.0)
AJCC stage	I	91 (68.9)
	II	20 (15.1)
	III	21 (16.0)
RT total dose (cGy)	Median (range)	6600 (4500–7020)
RT fraction size (cGy)	Median (range)	200 (180–210)
RT fraction number (fx)	Median (range)	33 (24–39)
Concurrent	No	81 (61.4)
chemotherapy	Xeloda + Cisplatin	10 (7.6)
regimen	Carboplatin + Paclitaxel	27 (20.4)
	Capecitabine/Xeloda only	10 (7.6)
	Others	4 (3.0)
Concurrent	No	81 (61.4)
chemotherapy	q 3 weeks	17 (12.9)
schedule	weekly	33 (25.0)
	others	1 (0.7)
Neoadjuvant chemotherapy	No	132 (100)

ECOG PS, European Cooperative Oncology Group Performance score; AJCC, American Joint Committee on Cancer; EG Junction, Esophagogastric junction; RT, radiotherapy.

**Table 2 cancers-15-05524-t002:** Acute and late toxicities.

No. of Patients, (%)					
Grade	0	1–2	3	4	5
*Acute toxicity*					
Esophagitis	35 (26.5)	94 (71.2)	3 (2.3)	0 (0)	0 (0)
Pneumonitis	125 (94.7)	7 (3.0)	0 (0)	0 (0)	0 (0)
Dermatitis	106 (80.3)	26 (19.7)	0 (0)	0 (0)	0 (0)
Anemia	122 (92.5)	8 (6.0)	2 (1.5)	0 (0)	0 (0)
Neutropenia	117 (88.7)	7 (5.3)	6 (4.5)	2 (1.5)	0 (0)
Lymphopenia	65 (49.3)	38 (28.7)	29 (22.0)	0 (0)	0 (0)
Thrombocytopenia	128 (97.0)	4 (3.0)	0 (0)	0 (0)	0 (0)
*Late toxicity*					
Pleural effusion	96 (72.7)	34 (25.8)	2 (1.5)	0 (0)	0 (0)
Pericardial effusion	115 (87.1)	15 (11.5)	1 (0.7)	1 (0.7)	0 (0)
Esophageal fistula	126 (95.4)	3(2.3)	2 (1.5)	1 (0.7)	0 (0)
Esophageal stenosis	109 (82.6)	17 (12.9)	6 (4.5)	0 (0)	0 (0)
Pneumonitis	132 (100.0)	0 (0)	0 (0)	0 (0)	0 (0)

## Data Availability

All datasets of the present study are available upon formal request from the corresponding author.

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
