# Peer review of "Treatment Outcomes of Proton Beam Therapy for Esophageal Squamous Cell Carcinoma at a Single Institute"

_cancers, 2023, doi:10.3390/cancers15235524_

Round 1

Reviewer 1 Report

Comments and Suggestions for Authors

This article is interesting, but there are some points to revise before publishing.

1.    You use “Proton Beam Therapy (PBT)” in Simple Summary session. “Proton Beam Therapy” is in uppercase, but I think it's in lowercase here.

2.    You said that “3-year 14 overall survival rates for patients with different stages of ESCC exhibited 81.0%, 62.9%, and 51.3%”. Did you mean stage 1-3? Could you revise Simple Summary for readers to understand easily?

3.    The template said that “We strongly encourage authors to use the following style of structured abstracts, but without headings:” about abstraction session. Could you check again?

4.    Could you tell us the detail methods of total dose definition? You said that “However, the strength of this single-institute study is the homogeneity in the treatment regimens, including dose-fractionation, radiation field, treatment planning, and delivery; therefore, the obtained results can be considered reliable” in Discussion section, but I do not understand your institutional rule. I think 45 CGE is too small for radical treatment.

5.    Could you tell us the definition methods whether patients with stage 1 receive concurrent chemotherapy or not, if you said that “However, the strength of this single-institute study is the homogeneity in the treatment regimens, including dose-fractionation, radiation field, treatment planning, and delivery; therefore, the obtained results can be considered reliable”?

6.    Could you tell us the detail definition of chemotherapy regimen, if you said that “However, the strength of this single-institute study is the homogeneity in the treatment regimens, including dose-fractionation, radiation field, treatment planning, and delivery; therefore, the obtained results can be considered reliable”?

7.    Could you show the definition of acute or late toxicity?

8.    I think that the detail data of salvage treatment. Could you show us?

9.    Some researchers said that we should not radical radiotherapy for EC after stenting. Did you do radical PBT for patients with esophageal stent?

Comments on the Quality of English Language

The English is reasonably good, but I think it needs to be revised again before publication.

Author Response

Thank you for your detailed review and thoughtful insights into my work. Please see the attachment.

Reviewer 2 Report

Comments and Suggestions for Authors

It is my pleasure to review the manuscript. The authors reported on “the effectiveness of Proton Beam Therapy  for esophageal squamous cell carcinoma”.  I think this is a novel review, but there are some corrections that need to be made.

Comments 

  1. Line 43
    As you pointed out, there are few reports of pure ESCC in western countries, but there are many reports in East Asia, so a correction is necessary.
  2. Line 87
    45 Gy is not a curative dose, but was it interrupted during the course of treatment? Please include the completion rate of treatment in the results.
  3. Line 93
    Please describe the extended field RT (EFRT) in detail.
  4. Line 101
    Please describe how the chemotherapy regimen was determined.
  5. ILine 132
    Is "66 CGE" typo?

Author Response

(The authors gave the same response as above.)

Reviewer 3 Report

Comments and Suggestions for Authors

Did you plan with SFO or MFO?

How did you perform setup verification? CBCT of X-ray? Did you use fiducials?

Did you perform CT during treatment to verify dose distribution or anatomy variation?

Did the median dose differ between tumor location?

Did you see lymphopenia during PT or after the end of the treatment? How can you distinguish lymphopenia from PT to lymphopenia from chemotherapy? 

Comments on the Quality of English Language

Good quality, only minor revision

Author Response

(The authors gave the same response as above.)
